# Database of the Italian disdrometer network

Elisa Adirosi[1], Federico Porcù[2], Mario Montopoli[1], Luca Baldini[1], Alessandro Bracci[1,2], Vincenzo Capozzi[3], Clizia Annella[3], Giorgio Budillon[3], Edoardo Bucchignani[4], Alessandra Lucia Zollo[4], Orietta Cazzuli[5], Giulio Camisani[5], Renzo Bechini[6], Roberto Cremonini[6], Andrea Antonini[7], Alberto Ortolani[7,8], Samantha Melani[7,8], Paolo Valisa[9], Simone Scapin[9]

[1] National Research Council of Italy, Institute of Atmospheric Sciences and Climate (CNR-ISAC), Rome, 00133, Italy
[2] Department of Physics and Astronomy "Augusto Righi", University of Bologna, Bologna, 40126, Italy
[3] Department of Science and Technology, University of Naples "Parthenope", Naples, 80143, Italy
[4] Meteorology Lab, Centro Italiano Ricerche Aerospaziali (CIRA), Capua, 81043, Italy
[5] Regional Agency for the Protection of the Environment of Lombardia (ARPA Lombardia), Milano, 20124, Italy
[6] Regional Agency for the Protection of the Environment of Piemonte (Arpa Piemonte), Torino, 10135, Italy
[7] Laboratory of Environmental Monitoring and Modelling for the sustainable development (LaMMA), Sesto Fiorentino (Florence), 50019, Italy
[8] National Research Council of Italy, Institute for the BioEconomy (CNR-IBE), Sesto Fiorentino (Florence), 50019, Italy
[9] Società Astronomica Schiaparelli, Centro Geofisico Prealpino, Varese, 21100, Italy

*Correspondence to*: Elisa Adirosi (elisa.adirosi@artov.isac.cnr.it)

**Abstract.** In 2021, a group of seven italian institutions decided to bring together their know-how, experience, and instruments for measuring the drop size distribution (DSD) of atmospheric precipitation giving birth to the Italian Group of Disdrometry (in Italian named: Gruppo Italiano Disdrometria, GID, https://www.gid-net.it/). GID has made freely available a database of 1-minute records of DSD collected by the disdrometer network along the Italian peninsula. At the time of writing, the disdrometer network is composed of eight laser disdrometers belonging to six different Italian institutions (including research centers, universities, and environmental regional agencies). This work aims to document the technical aspects of the Italian DSD database consisting of 1-minute sampling data from 2012 to 2021 in a uniform standard format defined within GID. Although not all the disdrometers have the same data record length, the DSD data collection effort is the first of its kind in Italy, and from here onwards, it opens new opportunities in the surface characterization of microphysical properties of precipitation in the perspective

of climate records and beyond. The Version 01 GID database can be downloaded here
https://doi.org/10.5281/zenodo.6875801 (Adirosi et al., 2022), while Version 02 can be downloaded here
https://doi.org/10.5281/zenodo.7708563 (Adirosi et al., 2023). The difference among the two versions is
the diameter-fall velocity relation used for the DSD computation.

## 1 Introduction

Disdrometers are punctual, non-captative devices able to measure the size and fall velocity (most of them)
of each single hydrometeor (solid or liquid) that falls into their measuring area which is at most 100 cm$^2$.
Just to have an order of magnitude, on average, 1 m$^3$ of air contains about 10$^3$ raindrops during
precipitation, including many more small drops than large ones (Uijlenhoet and Sempere Torres, 2006).
Particle size and fall velocity measurements allow computing the Particle Size Distribution (PSD) or the
Drop Size Distribution (DSD) in case of rain. Knowing the DSD, several rainfall parameters can be
obtained, such as rainfall rate, rainfall amount, radar reflectivity factor, liquid water content, and kinetic
energy of the falling particles.

Disdrometer data are useful for several applications that range from climatological, meteorological, and
hydrological uses to telecommunications, agriculture, and conservation of cultural heritage exposed to
precipitation. With respect to rain gauges, disdrometers provide more complete information about
precipitation, supplying not only the rainfall amount but also microphysical measurements. Considering
only the rainfall rate, disdrometers provide a huge improvement in detecting low intensity rain rate and
solid precipitation compared to rain gauge. In fact, tipping bucket rain gauges, among the most widely
used rain measuring devices, provide a measurement once a precipitation amount of 0.2 mm is collected
by the bucket, which prevents from estimating weak rainfall intensities over relatively short periods of
time. Estimation of solid precipitation from remote sensing and in-situ devices still represents a great
challenge due to the higher variability of shape, dimension, orientation, density, and habit of the solid
hydrometeors with respect to liquid precipitation. Microphysical information obtained by disdrometers
can improve both the quantitative estimation of solid precipitation (Capozzi et al., 2020; Bracci et al.,
2021) and the classification of precipitation types (Fehlmann et al., 2020).

An accurate characterization of the PSD (the DSD for rain) is useful for different applications such as:

- improve the accuracy of numerical weather prediction (NWP) models for precipitation forecasting (Van Den Heever and Cotton, 2004; Yang et al., 2019),

- increase the knowledge of the physical processes involved in the formation and evolution of precipitation, also considering the aerosol-hydrometeor interaction and the spatial variability at small scale (Tapiador et al., 2010; Tokay and Short, 1996; Bhupendra et al., 2021; Abbott and Cronin 2021),

- evaluate the effects of climate change on precipitation characteristics and intensity (Leinonen et al., 2012; Hachani et al., 2017),

- quantify the erosion effects of the precipitation on the soil and on the cultural heritage exposed to precipitation due to the kinetic energy of the hydrometeors (Kinnell, 2005; Serio et al., 2019),

- improve and validate the quantitative precipitation estimation (QPE) from remote sensing devices such as ground-based (Villarini, and Krajewski, 2010; Adirosi et al., 2018) and space-borne (Iguchi et al., 2009; Adirosi et al., 2021) weather radars,

- characterize the precipitation attenuation effects in microwave telecommunications to properly design the links and exploit these opportunistic attenuation signals to estimate precipitation (Giannetti et al., 2017; de Vos et al. 2019).

Disdrometers are classified according to their measurement principle: impact-type, infrared (laser or scatter), video, and radar type. To date, laser disdrometers are the most widely adopted for precipitation measurements, thanks to the good compromise between accuracy, purchase and installation cost, and low maintenance. The presented database is composed of data collected by laser disdrometers of two different manufacturers, namely OTT HydroMet GmbH, Kempten, Germany, and Thies Clima (Adolf Thies GmbH, Göttingen, Germany) that represent the overwhelming majority of the disdrometers used worldwide.

In general, disdrometric measurements are affected by several errors caused by: (i) statistical sampling, (ii) instrument limitations (i.e., resolution and sensitivity), and (iii) environmental factors such as wind effect, splashing or external interference from, e.g., insects or spider webs. Among the environmental factors, wind is recognized as the most significant source of measurement biases, and some studies have

been presented to mitigate its effects on disdrometers data (Friedrich et al., 2013; Capozzi et al., 2021).

Errors due to instrumental limitations depend on the type of disdrometer and the adopted measurement principle and can affect the measured DSD in different ways. Several authors have compared measurements of different disdrometers and have found systematic differences in the shape of measured drop spectra and corresponding integral parameters (e.g., Tokay et al., 2001; Krajewski et al., 2006; Thurai et al., 2011; Adirosi et al., 2020; Fehlmann et al., 2020).

Despite their potential role, disdrometers are not yet widely employed by meteorological and hydrological operational services, likely due to the lack of knowledge about their performance and accuracy in relation to environmental conditions, and missing standards for calibration, maintenance, and processing (Lanza et al., 2021). On the other hand, the use of disdrometers data for research purposes is increasing worldwide, and the first attempts to network these devices are emerging. One example is the UK

disdrometers network set up in 2017 for validating weather radar estimates (Pickering et al., 2019), or the Disdrometer and Radar Observations of Precipitation (DROP) network set up in 2010 as part of the Global Precipitation Measurement (GPM) Ground Validation (GV) program and still available around the NASA Wallops Flight Facility (Petersen et al., 2020). Another example is the network realised by the Italian Group of Disdrometry (in Italian: Gruppo Italiano Disdrometria, GID, https://www.gid-net.it/) here

presented. GID was set up  in 2021 thanks to a spontaneous collaboration of different Italian institutions (including research centers, universities, and environmental regional agencies) that manage disdrometers over the Italian peninsula.

The main aim of GID is to create a network between owners and users of disdrometers data in Italy in order to capitalize the instrumental resources and the available know-how, and to maximize the usefulness

of these precious measures in various fields of application. For these reasons, GID believes it is important to make freely available its own database that is composed of several years of 1-minute DSDs collected by 8 laser disdrometers along the Italian peninsula, processed with a common standard format defined within GID.

In section 2, a brief technical description of the laser disdrometers adopted in the GID network is provided

along with a detailed description of the network organization; Section 3 describes the common processing

adopted by GID to provide a uniform and accurate database of disdrometer data; Section 4 describes the GID database, and finally in Section 5, all necessary information on data access is reported.

## 2 Device and network description

In this section a brief description of the two types of laser disdrometer available in the GID network is provided.

### 2.1 Thies Clima Laser Precipitation Monitor

The Laser Precipitation Monitor (LPM) manufactured by Thies Clima (www.thiesclima.com), hereinafter TC, is a laser disdrometer and consists of a diode and optics which produces a parallel 780 nm laser beam with a detection area of $20 \times 228$ mm (45.6 cm$^2$). When the precipitation particle falls through the light beam, the received signal is reduced; the amplitude of the reduction is related to the size of the particle, and the duration of the reduction is related to the fall speed. The number of detected particles is recorded in a 22 size $\times$ 20 fall velocity matrix (although the first version of the TC recorded data in a $20 \times 20$ matrix). The particle diameter classes range between 0.125 mm and 8 mm, while the fall velocity ranges between 0.2 m s$^{-1}$ to 10 m s$^{-1}$. Lanzinger et al. (2006) and de Moraes Frasson et al. (2011) provided information regarding the factory calibration process and apparent accuracy of TC. The measurement uncertainty for the volume measurement under laboratory conditions is 2.2%.

### 2.2 OTT Parsivel 2

The OTT Parsivel2 (hereinafter P2) is a laser-based optical disdrometer to simultaneously measure PARticle SIze and VElocity of liquid and solid precipitation. P2 is an upgraded version of the OTT Parsivel (Löffler-Mang M., and Joss, J. 2000). The disdrometer has an optical sensor that produces a horizontal sheet of light (30 mm wide and 1 mm high, 180 mm long). In the receiver the light sheet is focused on a single photodiode. In clear sky conditions, the receiver produces a 5-V signal at the output of the sensor. Passing through the light sheet, particles partially block this light sheet causing a temporary reduction of the voltage. The reduction of the signal amplitude provides information on the size of the particle, while the reduction of the signal duration allows an estimation of the particle velocity. In

particular, the width of the maximum blocked area provides the maximum horizontal dimension of a drop, then the drop's equivalent diameter ($D_{eq}$) is computed assuming that a particle is a horizontally oriented oblate spheroid with axis ratio i) equal to 1 if $D_{eq} \leq 1$ mm, ii) varying linearly from 1 to 0.7 if 1 mm $< D_{eq}$ $< 5$ mm, and iii) equal to 0.7 for $D_{eq} \geq 5$ mm (Tokay et al., 2014). This assumption is reliable for rain, but
it is not appropriate for solid precipitation.

The raw output provided by the manufacturer's software, either at 10-second or 1-minute intervals, is the number of drops in 32 size and 32 fall velocity classes, with variable widths. The particle size ranges from 0.062 to 24.5 mm, while the fall velocity ranges from 0.05 to 20.8 m s$^{-1}$. However, the first two size classes, which correspond to diameters less than 0.2 mm, have been left empty by the manufacturer due
to the low signal-to-noise ratio. The Parsivel disdrometer was originally designed for the determination of radar reflectivity-rainfall relations, therefore its drop detection capability is lower in the left end of the drop spectrum (namely the small diameters). Indeed, this part of the spectrum has less influence on rain rate and radar reflectivity but may be important for cloud physics. In the Technical Data file available online on the OTT website (www.ott.com), the manufacturer reports that the measurement accuracy is $\pm$
1 size class up to 2 mm, and $\pm$ 0.5 size class for particles above 2 mm. In terms of rainfall rate for liquid precipitation this corresponds to an accuracy of $\pm 5\%$.

### 2.3 Disdrometers comparison in literature

Intercomparison between P2 and TC has been the subject of several published works. Upton and Brawn (2008) compared the DSD measured by the OTT Parsivel and the TC disdrometer. They found that TC
measures a higher number of drops respect to OTT Parsivel (i.e., Parsivel counts 74% of the drops recorded by TC in light rain, R<1 mm h-1, and 80% for R>1 mm h-1), however this difference in the drop count depends mainly to the fact that the TC measures about three times the number of small drops (D less than about 0.6 mm) recorded by the OTT Parsivel. Nevertheless, a good agreement between the precipitation amount recorded by a collocated rain gauge and the TC disdrometer was found (Upton and
Brawn 2008). Similar conclusions are reported by Angulo-Martínez et al. (2018), who compared two years of TC data with a collocated OTT Parsivel2 data. They found that TC recorded on average twice the number of particles than Parsivel2, but most of the differences are observed for very small drops. The

application of a filter criterion to the size-fall velocity matrix strongly reduces these discrepancies. In terms of rainfall rate, the TC is more sensitive to precipitation detection but overestimates rainfall amount with respect to Parsivel2.

Regarding the comparison of P2 with 2-Dimensional Video Disdrometer (2DVD), by Joanneum Research Forschungsgesellschaft mbH, Graz, Austria, which is considered the most accurate commercial disdrometer for DSD measurements, Tokay et al. (2016) and Park et al. (2017) found a very good agreement in the concentration of midsize drops (0.6–4.0 mm in diameter), and, as a consequence, in the rainfall rate for light and moderate precipitation, while in heavy rain P2 tends to overestimate large drops (likely due to binning effects). Finally, P2 detects a higher number of very small drops. Fehlmann et al. (2020) compared 2-year of TC data with the most accurate 2DVD and found that the number of particles with diameters between 0.5 and 3.5 mm is slightly underestimated by TC. Conversely, the number of the smaller and larger particles is overestimated, with the discrepancy for the larger drops (D >5 mm) being much higher than the one for the smaller drops.

Finally, Adirosi et al. (2018) found negligible differences in using TC, Parsivel2, or 2DVD drop size distributions to establish weather radar algorithms.

## 2.4 GID Network

At the time of writing this paper, the GID network is composed of 6 TC and 2 P2 located along the Italian peninsula, as shown in Figure 1, along with some pictures of the installed devices.

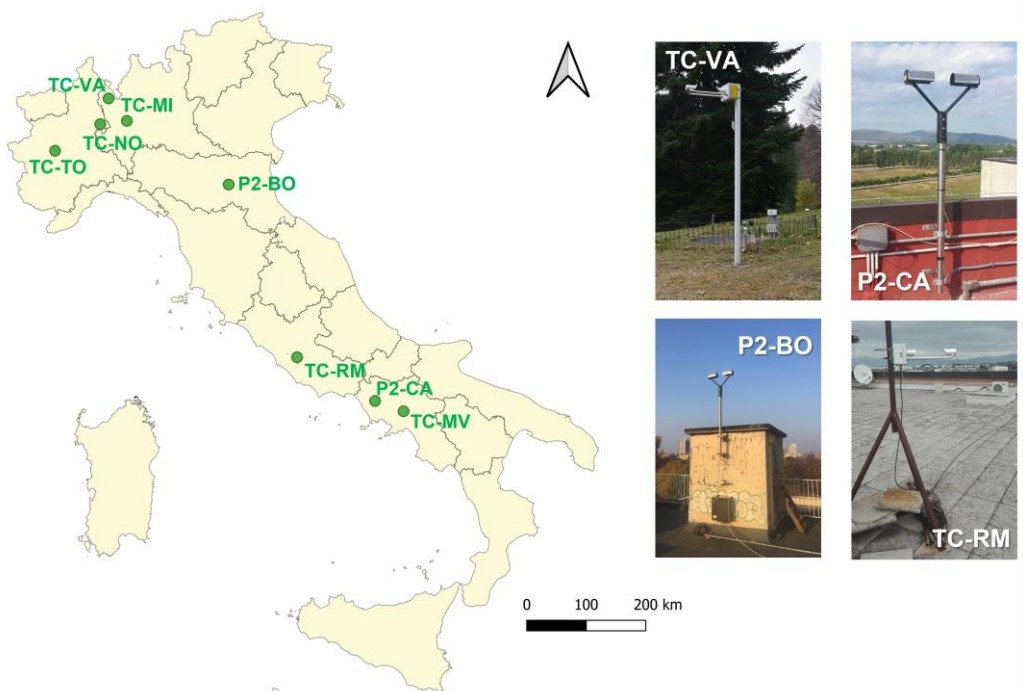

**Figure 1: Locations of the GID network disdrometers along with pictures of some installations. In the left panel, the prefix TC and P2 stand for Thies Clima and Parsivel 2 type disdrometer, respectively, whereas the suffixes indicate the locations: VA (Varese), MI (Milan), NO (Novara), TO (Turin), BO (Bologna) RM (Rome), CA (Capua) and MV (Montevergine).**

The geographical distribution of the disdrometers is not homogeneous along Italy (see Figure 1, left) due to the nature of the GID network. In fact, it was born thanks to the spontaneous collaboration of different Italian disdrometer owners without the possibility to decide the installation locations. However, one of the aims of GID is to enlarge the network with other disdrometers already available in Italy (if any) or with new devices. In the latter case the site identification will be driven by the goal of providing a more

homogeneous distribution of disdrometers, filling evident coverage gaps, especially in the South and in the two main Italian islands (i.e., Sicily and Sardinia).

The following Italian institutions are members of the GID network:

- National Research Council of Italy, Institute of Atmospheric Sciences and Climate (ISAC-CNR)
- National Research Council of Italy, Institute for the BioEconomy (IBE-CNR)
- Laboratory of Environmental Monitoring and Modelling for the sustainable development (LaMMA)
- Italian Aerospace Research Centre (CIRA), Meteorology Laboratory

- Department of Physics and Astronomy "Augusto Righi", University of Bologna (UniBo)
- Department of Science and Technology, University of Naples "Parthenope" (UniParth)
- Regional Agency for the Protection of the Environment of Lombardia (Arpa Lombardia)
- Regional Agency for the Protection of the Environment of Piemonte (Arpa Piemonte)
- Società Astronomica Schiaparelli, Centro Geofisico Prealpino (CGP)

Table 1 summarizes the primary information regarding the disdrometers of the GID network. In addition to the station's name (ID), Table 1 provides the location and coordinates of the installation site, the name

of the owner and the managing institutions, the month and year of the first measurement and the number of rainy minutes available until December 2021. The disdrometer coordinates are referred to the World Geodetic System-84 (WGS-84). The longest dataset is the one collected by TC-RM, which consists of almost 10 years of disdrometer data. However, it should be noted that some interruptions due to different causes can be present in the time series of the disdrometer measurements. Most of the devices are installed

in research facilities or measurement sites, co-located with other meteorological devices (such as raingauge, wind profiler, radar, visibilimeter, optical particle counter, etc.).

Except for the TC-MV and TC-VA, the disdrometers are located in relatively flat terrain (TC-MV is the only device located in a mountain environment). Furthermore, 4 disdrometers are located in urban areas and 4 in rural areas (namely TC-RM, TC-MV, P2-CA, TC-VA). The TC at Torino site is the old version

of the TC disdrometer. Following the Köppen–Geiger climate classification (Kottek et al., 2006), all the disdrometers are located in group C (temperate climate); however, the TC-MI, TC-TO, TC-VA, TC-NO, and P2-BO fall into the Csc (Mediterranean cold summer climates) area while the others in the Csa (Mediterranean hot summer climates) area.

| ID | Location | Latitude | Longitude | Height ASL (m) | First measurement | Owner | Manager | site classification | n. of rainy minutes |
|----|----------|----------|-----------|----------------|-------------------|-------|---------|---------------------|---------------------|
| TC-VA | Varese | 45.8316 | 8.7989 | 433 | April 2021 | CGP | CGP | rural, Csc | 77980 |
| TC-MI | Milano | 45.4904 | 9.1947 | 150 | April 2014 | ARPA Piemonte | ARPA Lombardia | urban, Csc | 42228 |

| | | | | | | | | | |
|---|---|---|---|---|---|---|---|---|---|
| TC-NO | Novara | 45.4402 | 8.6198 | 157 | August 2021 | ARPA Piemonte | ARPA Piemonte | urban, Csc | 11094 |
| TC-TO | Torino | 45.0294 | 7.6549 | 250 | January 2014 | ARPA Piemonte | ARPA Piemonte | urban, Csc | 253363 |
| P2-BO | Bologna | 44.4993 | 11.3538 | 65 | December 2018 | UniBo | UniBo | urban, Csc | 20252 |
| TC-RM | Roma | 41.8425 | 12.6464 | 102 | September 2012 | ARPA Piemonte | ISAC-CNR | rural, Csa | 209808 |
| P2-CA | Capua | 41.1192 | 14.1721 | 70 | July 2015 | CIRA | CIRA | rural, Csa | 106875 |
| TC-MV | Osservatorio di Montevergine | 40.9365 | 14.7291 | 1280 | December 2018 | UniParth | UniParth | rural, Csa | 60259 |


**Table 1: Information regarding the disdrometers of GID network. In the second-last column the site classification includes information on the surrounded area (i.e. urban or rural) and the Köppen–Geiger climate classification (Kottek et al., 2006), i.e. Csc (Mediterranean cold summer climates) or Csa (Mediterranean hot summer climates).**

## 3 Data Processing

The TC and P2 raw data consist of a 1-minute size-velocity matrix that contains the number of hydrometeors collected by the device for each drop size and fall velocity bin. The dimension of the matrix depends on the device: a 32×32 matrix for P2; a 22×20 matrix for TC (20×20 for the old version of TC). Knowing the size-velocity matrix, the DSD and the corresponding rainfall parameters can be obtained. However, to limit the differences between TC and P2 within reasonable limits (see the relative

considerations in Sec. 2.3) and to improve the accuracy of the obtained DSD and geophysical parameters, a filter criterion has been applied to the raw data. The latter is a common procedure, widely adopted in disdrometer-related studies.

The data processing adopted by GID and applied to all the disdrometers of the GID network is described below. The processing is valid only for liquid precipitation since an accurate estimation of PSD for mixed

or solid precipitation is more challenging and will be the main goal of further investigations. The selection of liquid precipitation samples was made by applying the fall velocity filter criterion described below.

When temperature data nearby the disdrometer are available, a further filtering criterion based on temperature is applied. The latter consists in eliminating the measured rainfall records with air temperature below 4°C.

Two different versions of the GID database are available on-line. The difference among them is the diameter-terminal fall velocity relation adopted in the processing. In particular, Version 02 of the GID data processing is composed of the following steps:

1.    Application of the fall velocity filter criterion to the 1-minute size-velocity matrix. The adopted criterion removes drops with a fall velocity outside the ±50% interval around the theoretical

diameter-fall velocity relation proposed by Atlas et al. (1973) and based on the observations of Gunn and Kinzer (1949). Furthermore, the Atlas et al. (1973) relation has been modified to take into account the terrain height of the disdrometer installation site (i.e., Foote and Du Toit, 1969; Porcù et al. 2014; Bringi and Thurai, 2005); therefore, the adopted fall speed is:

$$v_h(D) = v_o(D) \left(\frac{\rho_0}{\rho_h}\right)^{(0.375+0.025D)} = (9.65 - 10.3e^{-0.6D}) \left(\frac{\rho_0}{\rho_h}\right)^{(0.375+0.025D)} \tag{1}$$

where $h$ is the height (in m) of the site above the sea level (ASL), $v_0$ is the terminal fall speed at

sea level, and $\rho_0$ and $\rho_h$ (in kg/m³) are the air density at sea level and at height $h$, respectively. The values of the air density have been obtained assuming the International Standard Atmosphere Model (Bringi and Thurai, 2005). As example Figure 2 shows the filter mask for P2 and TC at sea level. Please note that this criterion is widely adopted in the literature (i.e., Thurai and Bringi 2005, Jaffrain and Berne 2011, Hauser 1984, Tokay et al. 2001, Adirosi et al. 2015, Adirosi et al.

2014 among others) and can be applied to any disdrometer raw data, as long as independent size and fall velocity data are available.

2.    Computation of the DSD. The following equation is used to compute the DSD only for 1-minute samples with at least 11 drops:

$$N^{P2;TC}(D_i) = \frac{1}{A^{P2;TC}\Delta t \, \Delta D_i^{P2;TC}} \sum_{j=1}^{C_v^{P2;TC}} \frac{n_{j,i}}{v_j} \tag{2}$$

where the superscript indicates the specific instrument, $N(D_i)$ is the drop size distribution (mm$^{-1}$ m$^{-3}$), $\Delta t$ is the sampling time (namely 60 s), $A$ is the instrumental measuring area (m$^2$), $v$ (m s$^{-1}$) is the theoretical fall velocity in Eq. (1), $\Delta D$ is the width of the size bin, $n_{j,i}$ is the number of drops measured in the $i$-th diameter class and $j$-th fall velocity class, and $C_v$ is the total number of fall velocity bins. The width of each diameter class is provided by the manufacturers.

3. Application of the rain/no-rain criterion. Knowing the DSD, the rainfall rate ($R$ in mm h$^{-1}$) can be easily computed as

$$R = 6\,\pi\,10^{-4} \sum_{D_{min}}^{D_{max}} v(D)N(D)D^3\,dD$$

(3)

A 1-minute sample is considered a rainy minute if $R > 0.1$ mm h$^{-1}$.

4. Data organization. Only the DSDs computed for the rainy minutes are saved. The data are stored in 1-year files named as "GIDVxx_ID_YEAR" where, Vxx indicates the version of the GID data processing, ID is the identification number of the disdrometer as shown in the first column of Table 1, and YEAR is the year when the data have been collected. Each file contains:

    a. Column 1: year

    b. Column 2: month

    c. Column 3: day

    d. Column 4: hour

    e. Column 5: minute

    f. Column 6 to end: DSD i.e., values of $N^{P2;TC}(D_i)$ in Eq. (1) for each bin $D_i$.

    Times are expressed in UTC.

With respect to Version 02, Version 01 of the GID algorithm does not apply the adjustment of the terminal fall velocity with respect to the height. In practice, Version 01 of the GID algorithm uses the Atlas et al. (1973) fall velocity at sea level for all the sites. For the highest GID site (i.e., TC-MV) the differences between $v_0(D)$ and $v_h(D)$ are -4.8% for $D = 0.1875$ mm and $-7.7\%$ for $D = 9$ mm. However, most of the GID disdrometers are located at low altitudes ($h < 400$ m ASL) where the effects of the site height can be considered negligible (i.e., less than 2%).

Figure 3 shows the normalized bias among $v_0(D)$ and $v_h(D)$ for different heights. The negative sign means
that $v_h(D)$ is higher than the one at sea level. The use of $v_h(D)$ has an impact also on the adopted fall
velocity mask. Comparing the TC-MV DSDs computed with V01 and V02 GID algorithm, we obtained
a mean normalized bias (NB) equal to -16%. However, the highest errors are found for the first diameter
class (D = 0.1875 mm) and for the last four (D > 6.75 mm), while for the other diameter classes the mean
difference is 6.7%. Finally, in terms of rainfall rate we obtained a NB = -3.2%.

Please note that in the GID database, only the DSDs are available. However, from the DSD data, other
DSD and rainfall parameters can be derived, such as: mass-weighted mean raindrop diameter ($D_m$), DSD
intercept parameter ($N_w$), rainfall rate ($R$), kinetic energy ($K$), liquid water content (LWC), and, assuming
a microphysical model and a scattering model for drops, relevant measurements for radar remote sensing
of precipitation, like radar reflectivity factor at horizontal polarization ($Z_h$), specific attenuation due to
rainfall ($k$), differential reflectivity ($Z_{dr}$), specific differential propagation phase shift ($K_{dp}$) and many
others (Bringi and Chandrasekar 2001).

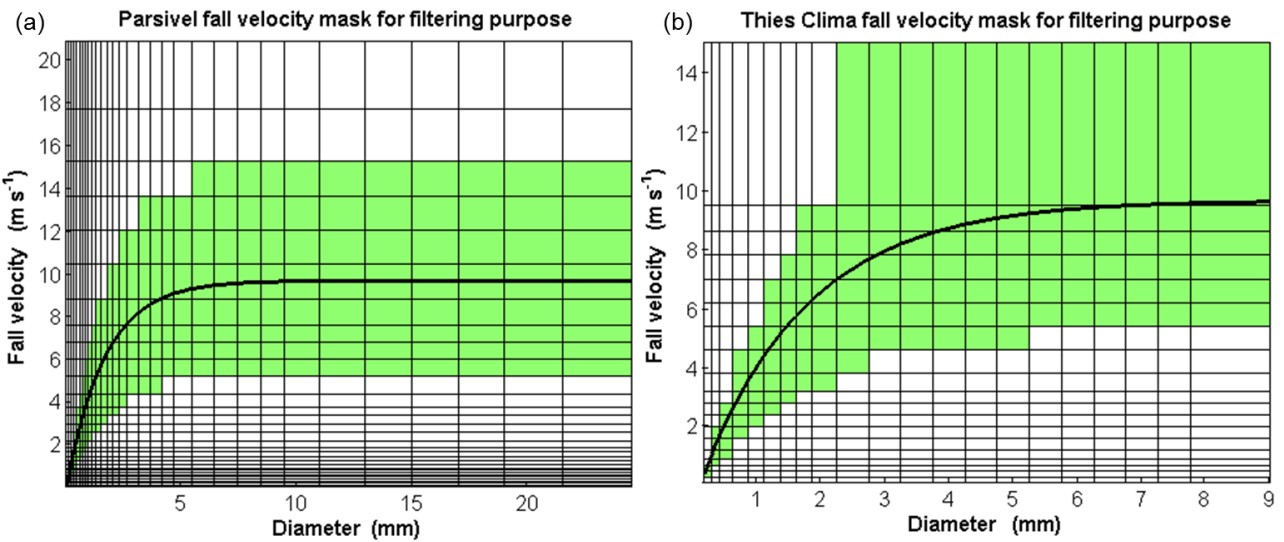

Figure 2: fall velocity masks for P2 (a) and TC (b) disdrometers. The data in the white bins are removed by the filtering criterion.

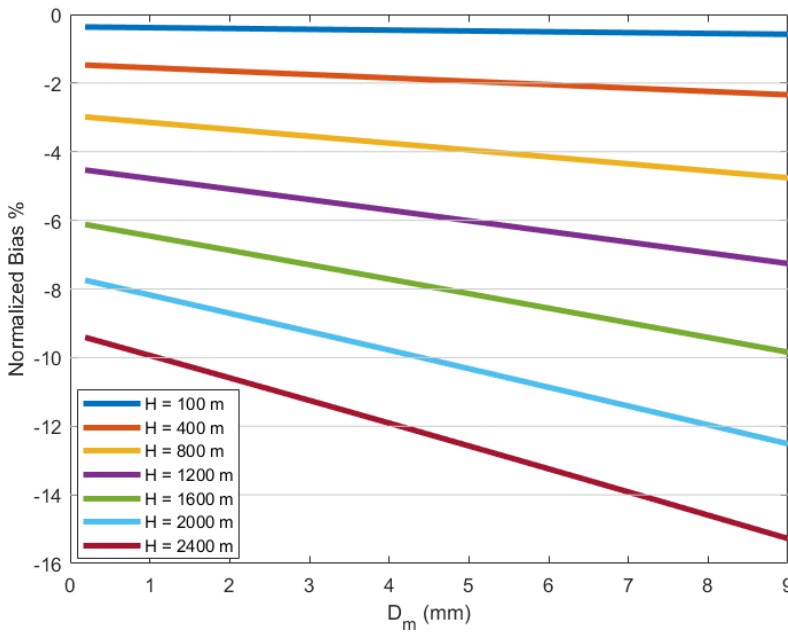


**Figure 3: normalized bias $v_0(D)$ and $v_h(D)$ for different heights.**

## 4 GID Database structure

The GID database is freely available as described in section 5 and is structured as detailed in section 3. It is composed of the DSDs collected by the eight laser disdrometers of the GID network during rainy

minutes. For each disdrometer, data are available from the first measurement (see Table 1) to 31 December 2021. All the GID disdrometers are still working and, in the future, the GID is planning to upgrade the published database yearly with new measurements/new sensors. The main folder of the GID database is "GID_database_untill_Dec2021 " that contains 8 sub-folders, one for each disdrometer of the GID network. The name of these subfolders is the disdrometer ID in 5 digits (for example "TC-RM"). In

each of these folders, there is one .xlsx file for each year of measurement. The latter file reports the time and the DSDs collected by the selected disdrometer during a given year. The name of the file follows the rule explained in section 3. For example, if the DSDs collected by the disdrometer in Rome during 2016 are needed, the path is the following: "GID_database_untill_Dec2021/TC-RM/" and the file name is "GIDV02_TC-RM_2016.xlsx" for the DSDs obtained with fall velocity adjusted for the height and

"GIDV01_TC-RM_2016.xlsx" for DSDs obtained without fall velocity adjustment. Furthermore, in each device sub-folders, there is one txt file named "read_me_ID.txt" (where ID stands for the ID of the disdrometer as shown in Table 1) in which the following metadata are reported:

- General information: station ID, latitude, longitude, and height ASL of the disdrometer, url for the data visualization and date of the first measurement;

- Technical information: disdrometer type, processing version, units of latitude, longitude height, and DSD, time standard, and time resolution;

- Reference: DOI of the database and how to cite it, DOI of the reference paper and how to cite it, name of the owner institution, and email of the contact person;

- Note: this section reports any useful information, such as interruptions due to technical issues or
changes in the disdrometer location.

Figure 4 shows the schematic structure of the GID database from the main folder to the file header. As an example, in the scheme only the "TC-NO" folder is open which contains the files named .txt and .xlsx. The header of the .xlsx file and the main sections of the .txt file are also reported. The scheme for the
other folders is identical except that several .xlsx files (i.e., a file for each year of measurements) may be present, depending on the selected site. Furthermore, the header of the .xlsx file for the P2 disdrometer differs slightly from the corresponding header of the TC disdrometer due to the higher number of size classes. For the P2 disdrometer the header is: {year, month, day, hour, min, ND1, ND2, ND3, ND4, ND5, ND6, ND7, ND8, ND9, ND10, ND11, ND12, ND13, ND14, ND15, ND16, ND17, ND18, ND19, ND20,
ND21, ND22, ND23, ND24, ND25, ND26, ND27, ND28, ND29, ND30, ND31, ND32}.

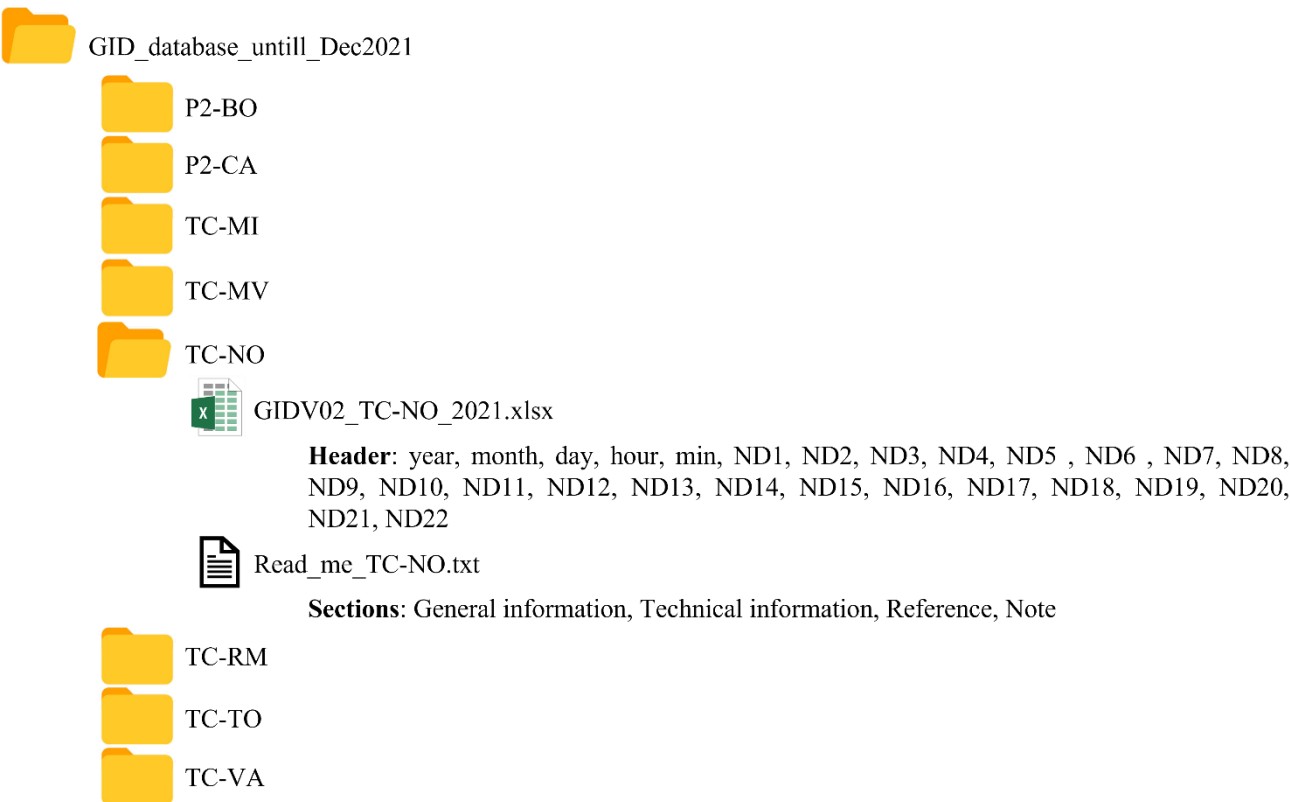

**Figure 4: schematic structure of the GID database.**

As an example, Figure 5 shows the DSDs collected by TC-RM around the precipitation peak (i.e., 138.08 mm h$^{-1}$) of an intense rainfall event on 14 February 2016. These DSDs are stored in the file called "GIDV02_TC-RM_2016.txt". The DSDs have the typical shape of natural DSD with a peak in the small diameter range (in this case around 0.5 mm). Knowing the DSD, the corresponding rainfall rate can be computed using Eq. (3). The maximum rain rate occurred at 22:56 UTC and the DSDs during and around this time have a quite high concentration of large drops, while at the beginning of the shown precipitation period (i.e., at 22:52 UTC), the rainfall rate was 2.9 mm h$^{-1}$ and the corresponding DSD in Figure 5 has a maximum drop diameter smaller than 3 mm.

Figure 6 shows, for each disdrometer of the GID network, the seasonal mean DSDs. With very few exceptions, the DSD shapes are very close for diameters smaller than 2 mm, while more differences are evident for mid-size and large diameters. In particular, the summer (JJA) DSD is the one with the highest concentration of mid-size and large diameters (i.e., likely due to the higher frequency of intense

convective rainfall events) while for winter (DJF), when stratiform precipitation is usually experienced, the DSD is the one with the smallest drop concentration; autumn (SON) and spring (MAM) DSDs are very close to each other and show intermediate values compared to the other two seasons.

Finally, Figure 7 shows the annual mean DSDs. For the disdrometers with more than one year of
measurements, the shapes of the DSDs are quite similar over the years, with differences mainly in the rightmost part of the spectrum (D > 5 mm). For example, the TC-RM dataset for 2017 shows the highest concentration of large drops (D > 5 mm) and 2016 the lowest concentration of large drops. P2-BO and TC-MV datasets for 2018 show smaller diameters compared to the other years, but the latter is due to the fact that data are not available for the whole year; in fact, both have been installed in December 2018.
The annual variability, in most of the sites, is less pronounced than the seasonal variability and is strictly linked to the natural variability of precipitation frequency and intensity.

Please note that the DSDs showed in Figure 5, Figure 6, and Figure 7 show a lower number of small drops (i.e. D < 0.5 mm) with respect to the ones reported in Thurai et al. 2019. The main reason for this difference is that in Thurai et al (2019) the DSDs were obtained combining data from conventional
disdrometer (that cannot capture the small drops, in particular the drizzle mode), and data form a high-resolution (50 microns) meteorological particle spectrometer, able to capture the small drops.

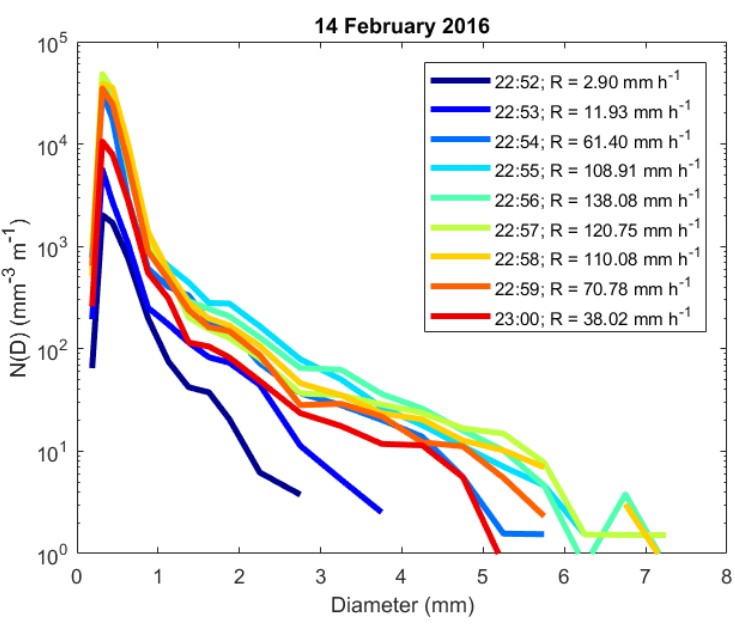

**Figure 5: example of DSDs collected by TC-RM. Each color-coded curve represents a different UTC times.**

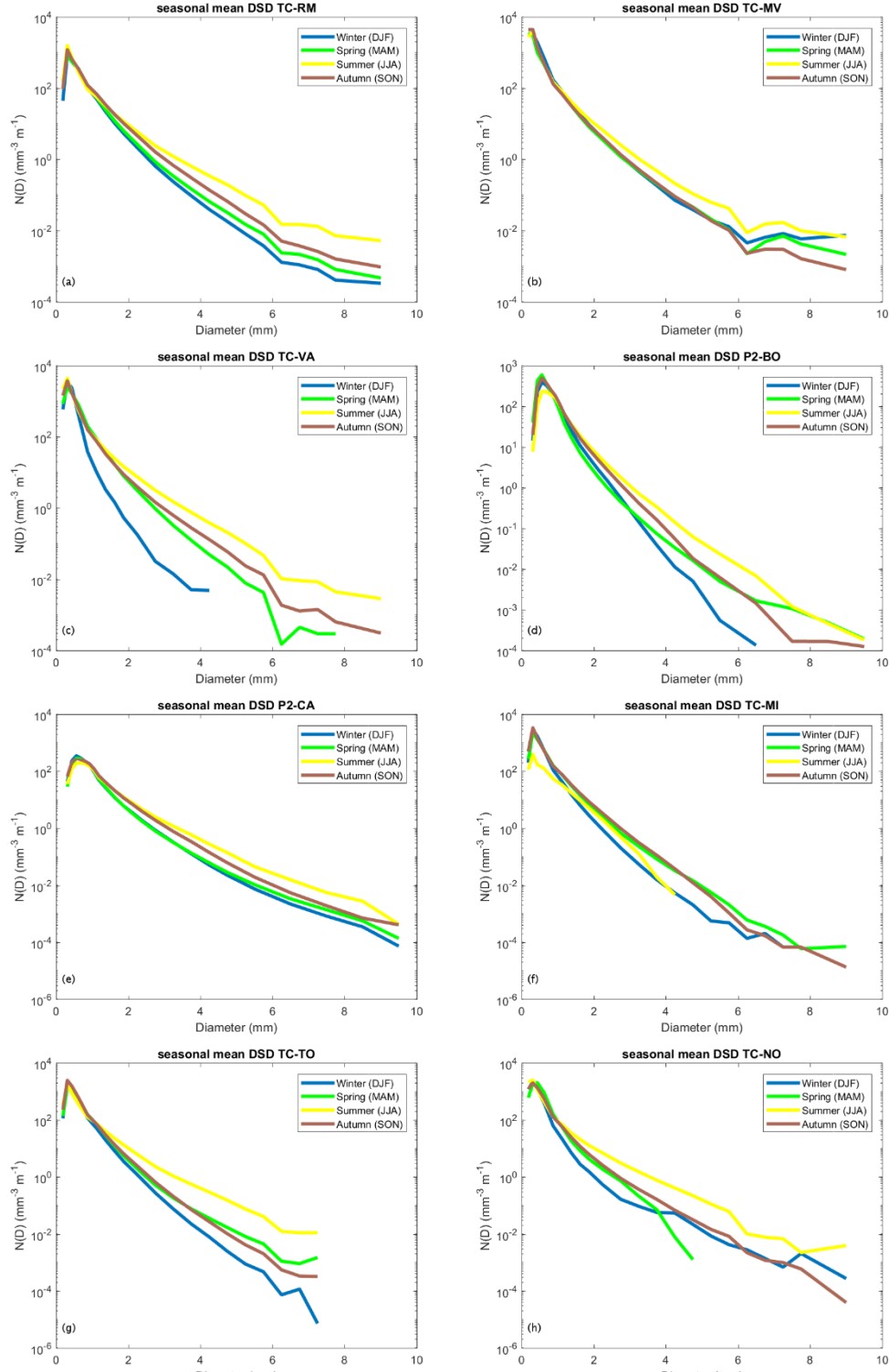

Figure 6: seasonal mean DSDs: (a) TC-RM, (b) TC-MV, (c) TC-VA, (d) P2-BO, (e) P2-CA, (f) TC-MI, (g) TC-TO, (h) TC-NO.

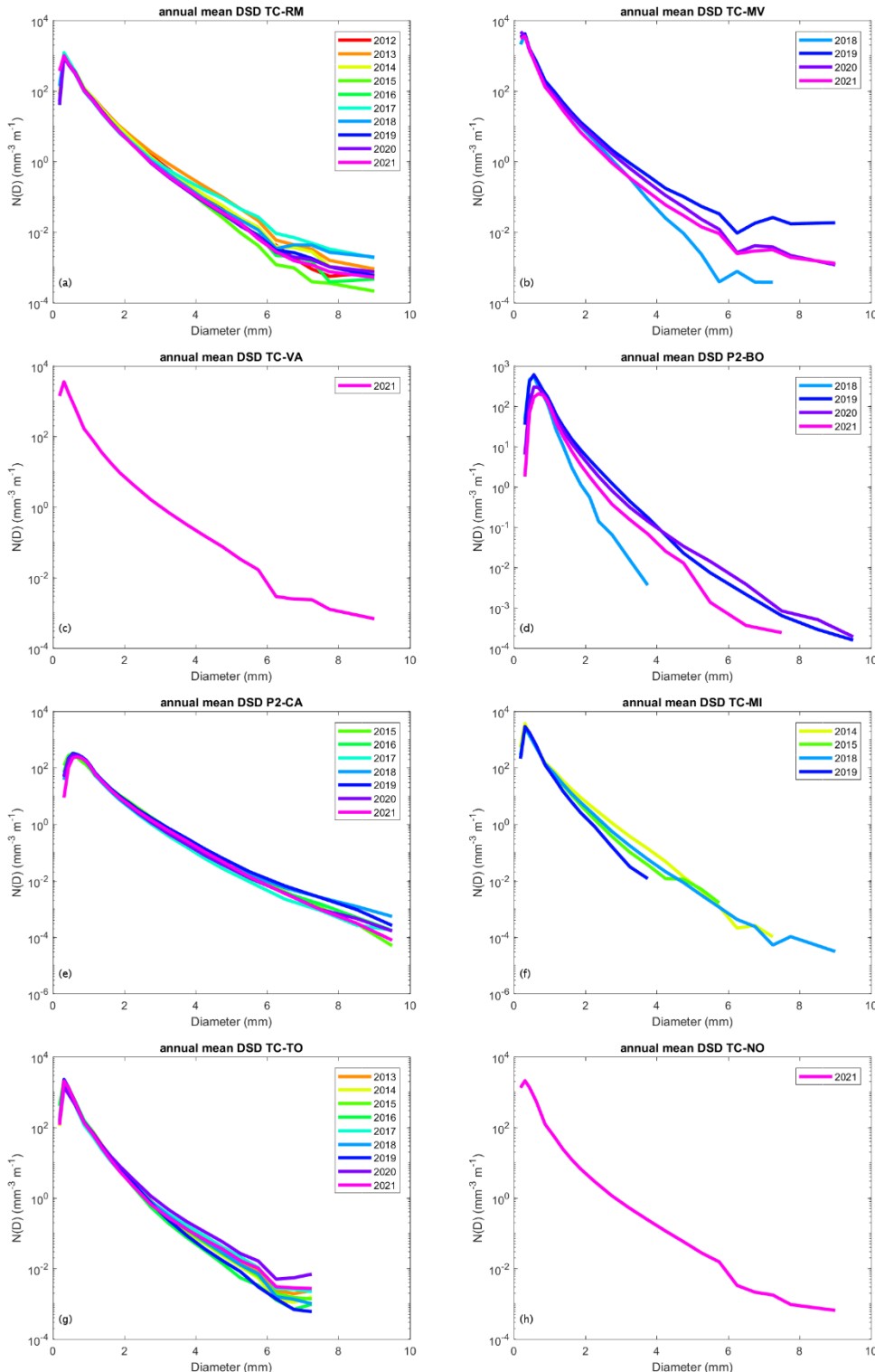

**Figure 7: annual mean DSDs: (a) TC-RM, (b) TC-MV, (c) TC-VA, (d) P2-BO, (e) P2-CA, (f) TC-MI, (g) TC-TO, (h) TC-NO.**

## 5 Data availability

One-minute DSDs obtained by processing the raw data collected by the GID network disdrometers are
available for free download under CC BY 4.0 license. The adopted processing has been described in
section 3, while the database structure is detailed in section 4. The GID database obtained with Version
01 of the algorithm is available at https://doi.org/10.5281/zenodo.6875801 (Adirosi et al., 2022). The
following citation should be used for every use of the data belonging to the GID database:

- Elisa Adirosi, Federico Porcù, Mario Montopoli, Luca Baldini, Alessandro Bracci, Vincenzo
Capozzi, Clizia Annella, Giorgio Budillon, Edoardo Bucchignani, Alessandra Lucia Zollo, Orietta
Cazzuli, Giulio Camisani, Renzo Bechini, Roberto Cremonini, Andrea Antonini, Alberto Ortolani,
Samantha Melani, Paolo Valisa, & Simone Scapin. (2022). Database of the Italian disdrometer
network (Version V01) [Data set]. Zenodo.

While the Version 02 of the GID database is available at https://doi.org/10.5281/zenodo.7708563 (Adirosi
et al., 2023) with the following citation:

- Elisa Adirosi, Federico Porcù, Mario Montopoli, Luca Baldini, Alessandro Bracci, Vincenzo
Capozzi, Clizia Annella, Giorgio Budillon, Edoardo Bucchignani, Alessandra Lucia Zollo, Orietta
Cazzuli, Giulio Camisani, Renzo Bechini, Roberto Cremonini, Andrea Antonini, Alberto Ortolani,
Samantha Melani, Paolo Valisa, & Simone Scapin. (2023). Database of the Italian disdrometer
network (V02) (Version V02) [Data set]. Zenodo.

These disdrometers are still collecting data and regular updates of their status along with updates of the
GID network are provided through the GID web site (www.gid-net.it). Furthermore, the raw data of the
GID network disdrometers can be provided under specific agreement. If interested in the raw data of a
specific disdrometer, please contact the reference person listed in the Read-me.txt file while if the raw
data of the whole GID database is of interest please email to gid.info@gid-net.it.

## 6 Conclusion

In this work, a centralizing effort of drop size distribution measurements is described for the Italian
territory. The result is the set-up of a spontaneous entity named GID (Italian Group of Disdrometry). GID,

so far, has gathered eight disdrometers over the Italian peninsula and centralized the data acquisition on a yearly basis. More importantly, the centralized data are stored on a public database and made freely available. However, at the time of writing this paper the procedure is not automatic, therefore some delay in the centralized collection and online publication of the DSDs is possible. Anyone interested in data that are not available online, can contact the GID (gid.info@gid-net.it) or the specific referent of the disdrometer. As an upgrade in the future, we are planning to exploit modern ICT methods to automate the whole process, from data collection to processing and public sharing on the online platform. With this initiative we hope to stimulate the national and regional weather services, and in general all the stakeholders (e.g., in the hydro-meteorological sector), to invest in the enhancement of existing and future disdrometer networks. Such strategy would be relatively cost-effective and will provide new insights into the microphysical properties of precipitation on a national scale, thus opening to a plenty of new applications and enhancing the accuracy of ground precipitation estimates. This could be relevant for a proper management of territory (from mitigation to risk) as well as for providing important feedback in the understanding of atmospheric processes and how these are strictly interlinked to a changing climate.

**Acknowledgements**

The Department of Science and Technology of the University of Naples "Parthenope" and the authors of this work are grateful to the Benedectine Community of Montevergine Abbey for affording the opportunity to install the laser-optical disdrometer on the Montevergine observatory terrace. The instrument P2-BO (University of Bologna) was purchased within the OPERANDUM project, funded by the EU's H2020 research and innovation programme (Grant Agreement No: 776848).

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
