# Peer review of "Database of the Italian disdrometer network"

_Earth System Science Data, 2022_

## Author Comment (AC1)

**We thank the Reviewer for the time spent on our manuscript and for the comments. Answers to the comments are in red.**

This paper presents the technical aspects of the new freely available drop size distribution (DSD) database in Italy contributed by the disdrometer network from the corporation of seven Italian institutions, namely the Italian Group of Disdrometry (GID). This paper documented the technical details of the two types of laser disdrometers in the GID, six Thies Clima Laser Precipitation Monitor and two OTT Parsivel 2. The raw data was filtered by the fall velocity criterion to the 1-minute size-velocity matrix before computing of the DSD, and further filtered by the rain/no-rain criterion. The data was stored and shared in yearly XLSX files. The work documented in this paper does contribute to the frontier in the field of precipitation measurements, and promotes the expansion of the disdrometer network.

**Comments:**

1. The data was shared through webpage, https://doi.org/10.5281/zenodo.6875801. The most up-to-date data was in year 2021, which means data in this year 2022 is yet available. It is suggested to update the data on more frequently.

The procedure to download the data from the device, process them and upload the DSD on Zenodo repository is not automatic. The strong ICT effort needed to make this procedure automatic is beyond our capabilities at the moment and right now we are a bit far from that goal. However, we are committed to update the database yearly. Therefore, in the first months of 2023 the data of 2022 should be available on-line. Furthermore, anyone interested in our data can contact the GID (gid.info@gid-net.it.) or the referent of a single disdrometer to obtain disdrometer data with a much higher frequency. We will be more than glad to share our recent data. We add the following sentence in the text to explain the latter:

**"Anyone interested in data that are not available online, can contact the GID (gid.info@gid-net.it) or the referent of each disdrometer to have these data."**

2. It is also not very clear how the data sharing workflow is organized. It looks like we have some DSD data from eight disdrometers shared online, but not sure whether the data will be update in the future and if there is any delay for the latest data to be published online.

The paper refers to the data available since December 2021, although the DSDs collected after that date are available. However, to obtain these data a specific request to the GID or to the referent of the disdrometer is needed because right now we do not have the structure to organize and automatic download, process and online publication of the data.

3. In future work, I suggest using modern ICT methods to enable the automation and reduce the time delay in the data collection, data transferring, data processing and data sharing.

We completely agree with the Reviewer, and we add in the conclusion section this sentence at that regard:

"However, right now the procedure is not automatic, therefore some delay in the centralized collection and online publication of the DSDs is possible. As upgrade in the future, we are planning to exploit modern ICT methods to automatize the download of

the data from the disdrometer, process them and share on online platform. The current goal of the dataset is to make available to the science community a quite large dataset of data collected in Italy, processed with a common procedure through a single repository otherwise not so easily accessible."

---

## Author Comment (AC2)

We thank the Reviewer for the time spent on our manuscript and for the comments. Answers to the comments are in red.

The manuscript describes a new database of precipitation particle size measurements across Italy. These measurements are important since they provide insights about the type of precipitation, its microphysical origin, hydrologic impact, and enable calibration of remote sensing measurements and communication links. Although the manuscript provides some scientific analysis of this dataset, it fails to provide much detail or discussion of the results and as such it is not very useful for gaining new scientific insights about precipitation. Instead, the manuscript is more akin to an algorithm theoretical basis document in some regards, or serves as simply a means to document the new database.

The main goal of Earth System Science Data (ESSD) journal is the publication of articles on original research data (sets), furthering the reuse of high-quality data of benefit to Earth system sciences. For these reasons in our paper we focus more on the description of the new dataset, in terms of how it has been collected and organized, than on the scientific analysis of the data. We add only few plots to showcase our dataset and its potential and we hope that the database will be extensively used by end users (either in the research or private sectors) for scientific analysis and research.

Here are some major concerns with the manuscript:

- The manuscript lacks many key references related to existing disdrometer networks, instrument, and DSD studies.

    o The background needs to include a few more references to existing disdrometer networks. The GPM Ground Validation (GV) Program (Petersen et al. 2020) has operated the Disdrometer and Radar Observations of Precipitation (DROP) Facility, which consists of a network of video and laser disdrometer that have been deployed to GPM-related GV activites since 2010. The DROP Facility continues to operate around the NASA Wallops Flight Facility. This vast dataset of particle size distribution measurements is archived at NASA's GHRC DAAC (https://ghrc.nsstc.nasa.gov/home/field-campaigns/gpmgv).

    Petersen, W.A., Kirstetter, PE., Wang, J., Wolff, D.B., Tokay, A. (2020). The GPM Ground Validation Program. In: Levizzani, V., Kidd, C., Kirschbaum, D., Kummerow, C., Nakamura, K., Turk, F. (eds) Satellite Precipitation Measurement. Advances in Global Change Research, vol 69. Springer, Cham. https://doi.org/10.1007/978-3-030-35798-6_2

    We added this reference.

o  Need to cite Loffler-Mang and Joss (2000) since that is the first study on the Parsivel disdrometer.

   Löffler-Mang, M., and Joss, J. (2000). An Optical Disdrometer for Measuring Size and Velocity of Hydrometeors. Journal of Atmospheric and Oceanic Technology 17, 2, 130-139, available from: < https://doi.org/10.1175/1520-0426(2000)017<0130:AODFMS>2.0.CO;2>

   We added this reference.

o  In reference to computing radar reflectivity factor from Parsivel measurments, iite Loffler-Mang and Blahak (2001).

   Löffler-Mang, M., and Blahak, U. (2001). Estimation of the Equivalent Radar Reflectivity Factor from Measured Snow Size Spectra. Journal of Applied Meteorology 40, 4, 843-849, available from: < https://doi.org/10.1175/1520-0450(2001)040<0843:EOTERR>2.0.CO;2>

   Since this reference regards snow size spectra while we use only rain DSDs we believe that this reference is not necessary.

o  Add source of reported Parsivel measurement accuarcy numbers provided in Section 2.2

   We added the source of these information.

o  Give examples of studies that use filtering when analyzing disdrometer measurements.

   We added several studies (with references) that applied the same filtering criterion to disdrometer data.

o  Section 3: Provide examples of studies that use DSD measurements to compute these additional integral parameters like LWC, Ze, etc.

   Done.

- The writing is very good, but the English grammar could be improved. It would be beneficial to have the next revision reviewed by a primarily English speaking proof-reading service before resubmitting.

   We carefully read the manuscript in order to improve the English grammar.

- This study uses the Gunn-Kinzer terminal velocity reference, which were obtained for mean sea-level. Foote and Du Toit (1969) have demonstrated that density affects the terminal fallspeed of raindrops. Hence, there is a need to correct the

fall-speed measurements for altitude, in particular the TC-MV site. It may also need to be done for the other sites (e.g., Thurai and Bringi corrected the terminal velocity computed from Atlas 1973 for a disdrometer located at an altitude of only 480-m ASL).

Foote, G. B., and Du Toit, P. S. (1969). Terminal Velocity of Raindrops Aloft. Journal of Applied Meteorology and Climatology 8, 2, 249-253, available from: < https://doi.org/10.1175/1520-0450(1969)008<0249:TVORA>2.0.CO;2>

Thurai, M., and Bringi, V. N. (2005). Drop Axis Ratios from a 2D Video Disdrometer. Journal of Atmospheric and Oceanic Technology 22, 7, 966-978, available from: < https://doi.org/10.1175/JTECH1767.1>

We thank the Reviewer for this comment. After some analysis we decided to reprocess all the datasets using the terminal fall velocity relation corrected for the height as in Thurai and Bringi (2005). Once done we will upload on Zenodo the new version of the database (Version 02) and add on the manuscript the corresponding doi. The figures of the revised manuscript are being updated considering the new version of the GID database. We are also adding new text in the revised manuscript at that regard.

*"Two different version of the GID database are available on-line. The difference among them is the diameter-terminal fall velocity relation adopted in the processing."*

*"Furthermore, the Atlas et al. (1973) relation has been adjusted to take into account the terrain height of the location at which the disdrometer is installed (i.e. Foote and Du Toit, 1969; Porcù et al. 2014; Bringi and Thurai, 2005). Considering the correction, the adopted fall speed is:*

$$v_h(D) = v_o(D)\left(\frac{\rho_0}{\rho_h}\right)^{(0.375+0.025D)} = (9.65 - 10.3e^{-0.6D})\left(\frac{\rho_0}{\rho_h}\right)^{(0.375+0.025D)}$$

(1)

*where h is the height (in m) above the sea level of the site and $\rho_0$ and $\rho_h$ (in kg/m3) are respectively the air density at sea level and at height h. The values of the air density have been obtained assuming the International Standard Atmosphere Model (Bringi and Thurai, 2005)."*

*"With respect to Version 02, the Version 01 of the GID algorithm does not apply the adjustment of the terminal fall velocity with respect to the height. In practice, Version 01 of the GID algorithm uses for all the sites the Atlas et al. (1973) fall velocity at sea level. For the highest GID site (i.e. TC-MV) the differences between $v_0(D)$ and $v_h(D)$ are -4.8% for D = 0.1875 mm and –7.7% for D = 9 mm. However, most of the GID disdrometers are located at low altitudes (h < 400 m ASL) where the error can be considered negligible (i.e. less than 2%). Figure 2 shows the normalized bias among $v_0(D)$ and $v_h(D)$ for different heights. The negative sign means that $v_h(D)$ is higher than the one at sea level. The use of $v_h(D)$ has an impact also on the adopted fall velocity mask. Comparing TC-MV DSDs computed with V01 and V02 GID algorithm, we*

*obtained a mean normalized bias (NB) equal to -16%. However, the highest errors are found for the first diameter class (D = 0.1875 mm) and for the last four (D > 6.75 mm), while for the other diameter classes the mean difference is 6.7%. Finally, in terms of rainfall rate we obtained a NB = -3.2%."*

- Need to include reason(s) why the TC and P2 disdrometer plots in Figures 4-6 show lower concentrations at the smallest drop diameters. The recent raindrop size measurements by Thurai et al. (2019) that use a disdrometer capable of better resolving the small diameter part of the spectrum is a good example.

  Thurai M, Bringi V, Gatlin PN, Petersen WA, Wingo MT. Measurements and Modeling of the Full Rain Drop Size Distribution. Atmosphere. 2019; 10(1):39. https://doi.org/10.3390/atmos10010039

The DSDs showed in **Errore. L'origine riferimento non è stata trovata.**, **Errore. L'origine riferimento non è stata trovata.**, and **Errore. L'origine riferimento non è stata trovata.** shown a lower number of small drops (i.e. D < 0.5 mm) with respect to the one reported in Thurai et al. 2019. The main reason of this difference is that in Thurai et al (2019) the DSDs have been obtained combining data from conventional disdrometer (that cannot capture the small drop end, in particular the drizzle mode), and data form a high-resolution (50 microns) meteorological particle spectrometer, able to capture the small drops. We added the following sentences in the revised version of the manuscript to clarify this point:

*"Please note that the DSDs showed in **Errore. L'origine riferimento non è stata trovata.**, **Errore. L'origine riferimento non è stata trovata.**, and **Errore. L'origine riferimento non è stata trovata.** shown a lower number of small drops (i.e. D < 0.5 mm) with respect to the one reported in Thurai et al. 2019. The main reason of this difference is that in Thurai et al (2019) the DSDs were obtained by combining data from conventional disdrometer (that cannot capture the small drop end, in particular the drizzle mode), and data form a high-resolution (50 microns) meteorological particle spectrometer, able to capture the small drops."*

- Include the number of DSD spectra for each site (e.g., in Table 1) since those are needed to assess statistical significance of climatological results in Figure 5.

  Done.

- What are possible reasons for the seasonal variability exhibited in the DSD results shown in Figures 5 and 6?

We added the following considerations in the revised manuscript.

*"In particular, the summer DSDs exhibit the highest concentration of mid-size and large diameters **(i.e. likely due to the higher frequency of intense convective rainfall events),** while in winter, **when stratiform precipitation is more frequently***

*experienced, the DSDs present the smallest concentration; autumn and spring DSD are very close with intermediate values with respect the other two seasons."*

Minor comments:

- Several mentions of the "old version TC" are in the manuscript. Please clearly state which site(s) has or had this version.

  Done.

- The Parsivel software computes the spherical volume-equivalent diameter (Deq) based on the measured particle diameter. The manuscript words this in a confusing manner that mentions the particle axis ratio (lines 153-155).

  We modified the sentence as follow:

  *"In particular, the width of the maximum blocked area provides the maximum horizontal dimension of a drops, than the drops equivalent diameter (Deq) is computed assuming that a particle is horizontally oriented oblate spheroid with axis ratio i) equal to 1 if Deq ≤ 1 mm, ii) that vary linearly from 1 to 0.7 if 1 mm < Deq < 5 mm, and iii) equal to 0.7 for Deq ≥ 5 mm (Tokay et al., 2014)."*

- Line 299: Spelling error..."expect" should be "except"

  Done.

- Suggest including in Figure 4 the rainfall rate for each 1-min DSD (e.g., another entry in the legend)

Done.